# Global Evidence of Aerosol-induced Invigoration in Marine Cumulus Cloud

Alyson Douglas[1] and Tristan L'Ecuyer[2]

[1]Atmospheric, Oceanic, and Planetary Physics Department, University of Oxford, Sherrington Rd, Oxford OX1 3PU
[2]Atmospheric and Oceanic Sciences Department, University of Wisconsin-Madison 1225 W Dayton St, Madison, WI 53706

**Correspondence:** Alyson Douglas (alyson.douglas@physics.ox.ac.uk)

**Abstract.** Aerosol-cloud-precipitation interactions can lead to a myriad of responses within shallow cumulus clouds including an invigoration response, whereby aerosol loading results in a higher rain rate, more turbulence, and deepening of the cloud layer. However few global studies have found direct evidence that invigoration occurs. The few satellite based studies that report evidence for such effects generally focus on only the deepening response. Here, we show evidence of invigoration beyond a deepening response by investigating the effects of aerosol loading on the latent heating and vertical motion profiles of warm rain. Using latent heating and vertical motion profiles derived from CloudSat radar observations, we show precipitating cumulus clouds in unstable, polluted environments exhibit a marked increase in precipitation formation rates and cloud top entrainment rates. However, invigoration is only discernible when the stability of the boundary layer is explicitly accounted for in the analysis. Without this environmental constraint, the mean polluted and pristine cloud responses are indiscernible from each other due to offsetting cloud responses in stable and unstable environments. Invigoration, or suppression depending on the environment, may induce possible feedbacks in both stable and unstable conditions that could subdue or enhance these effects, respectively. The strength of the invigoration response is found to additionally depend on cloud organization defined here by the size of the warm rain system. These results suggest that warm cloud parameterizations must account for not only the possibility of aerosol-induced cloud invigoration, but also the dependence of this invigorated state on the environment and the organization of the rain system.

## 1 Introduction

Aerosol-cloud interactions remain one of the largest sources of uncertainty in future climate projections (Boucher et al., 2013). Further, their role in climate feedbacks, particularly how they affect low clouds, controls the magnitude of the climate sensitivity (Zelinka et al.). However, despite the importance of tropical low clouds to the global climate, understanding their response to anthropogenic activity, including aerosol loading, remains a challenge (Bony and Dufresne, 2005). In particular, invigoration, or the enhanced size, depth, precipitation rate, or turbulence, of low clouds was hypothesized as a potential outcome of aerosol-cloud interactions decades ago but remains relatively unconfirmed from observations (Pincus and Baker, 1994; Rosenfeld et al., 2008). If invigoration of warm cloud precipitation occurs, it not only affects where and how much clouds precipitate, but the entire hydrological cycle (Li et al., 2011). Invigoration of warm cloud structure also has the potential to alter deep convection,

making eventual storms more intense and turbulent (Chen et al., 2017). Further, unlike studies that focus on the suppression of drizzle in shallow warm clouds, such as Ackerman et al. (2004) which found increased turbulence through suppression of drizzle by aerosol, herein we evaluate the effects of aerosol on warm rain events and define invigoration beyond just an increased in turbulence or vertical motion, but by changes in the latent heating structure throughout the cloud layer.

Previous studies focused on detecting warm cloud deepening as a signal of invigoration, as it theoretically implies increased turbulence and precipitation within the cloud (Altaratz et al., 2014). L'Ecuyer et al. (2009) showed warm, polluted precipitating clouds grow deeper than those in more pristine aerosol environments, those with minimal anthropogenic aerosol emissions, using space-borne radar observations from CloudSat. Christensen and Stephens (2011) similarly found as ships passed below marine stratocumulus, the locally affected clouds, as identified using a combination of radar and passive satellites, deepened. Yuan et al. (2011) found evidence of cloud deepening in trade cumulus when interacting with nearby volcanic emissions while Kubar et al. (2009) found evidence of increased liquid water amounts in highly polluted environments when controlling for cloud top height in all warm cloud types. On the other hand, Dey et al. (2012) found no evidence of cloud deepening in the smallest clouds studied, only an increase in extent. Never-the-less, while numerous observational studies have been able to discern a cloud depth response using both passive and active sensors, few have controlled for the environment in their estimates in order to certify that this response is due to aerosol-forced invigoration and not a confounding environmental signal. Additionally, a deepening cloud does not conclusively establish the physical processes associated with invigoration, such as increased turbulence and precipitation formation rate, only that another cloud adjustment process occurs.

Modeling efforts have proven more promising in, at least hypothetically, demonstrating invigoration of warm clouds is possible by aerosol. Recently, Wu and Chen (2021) found in the Weather Research and Forecast model an increase in drizzle rates in the most polluted runs while simulating north Pacific warm clouds. Dagan et al. (2017) found that as clouds reach an equilibrium state, the polluted scenarios are likely to see an increase in rain production due to enhanced instability. Capturing an increase in precipitation formation in simulated cloud environments under high aerosol conditions depends on the time the model is allowed to run, as aerosol loading increases the time to precipitation (Seifert et al., 2015). Precipitation suppression can also alter which type of clouds may eventually rain by altering water vapor transport, resulting in higher rain rates in regions downstream of the original suppression (Dagan and Chemke, 2016). Heiblum et al. (2019) used a LES model to show that clouds formed in higher aerosol environments release more latent heat and promote a larger rain cell size. Jiang et al. (2009) similarly used a LES and found clouds in polluted environments produced more evaporation at the cloud edge in simulated trade cumuli, producing more vertical motion. (Spill et al., 2019) found shallow convective clouds in high aerosol loading environments are more likely to deepen with a variable response of the rain rate. Clouds formed in polluted environments may experience an increase in droplet mobility, the amount of motion by each droplet not forced by gravity, which delays collision coalescence and changes the organization of liquid water within the cloud to a more invigorated state reaffirming Albrecht's original theory of a second aerosol indirect effect (Koren et al., 2015; Albrecht, 1989; Berg et al., 2008). (Seifert et al., 2015) saw a decrease in cloud lifetime with increasing droplet concentration due to desiccation of warm cloud cover from precipitation induced environmental feedbacks. Depending on the environmental conditions, the liquid water path of the cloud may decrease, signaling a curtailment, not invigoration, response (Jiang et al., 2006).

The environment plays a strong role in modulating warm rain processes and therefore must be considered when using observations to imply aerosol-forced invigoration of warm clouds (Stevens and Feingold, 2009). Prior work has shown that the environment controls the amount of suppression of precipitation within the cloud, which may modulate the amount of invigoration (L'Ecuyer et al., 2009). The strength of the marine boundary layer inversion controls cloud top height in many warm clouds (Wood, 2012) and has been shown to heavily influence rain formation rates in warm clouds (Nelson and L'Ecuyer,

2018). The magnitude and sign of warm cloud aerosol-cloud interactions is likewise heavily modulated by both the inversion strength and free atmospheric relative humidity (Douglas and L'Ecuyer, 2019). The humidity of the free atmosphere affects how aerosol impacts the distribution of liquid water throughout the cloud layer due to entrainment processes (Ackerman et al., 2004; Eastman and Wood, 2018). Both are considered within this study in order to constrain these confounding factors.

To a first order, the liquid water path controls the probability of a cloud raining (L'Ecuyer et al., 2009; Berg et al., 2006).

Aerosols, in turn, impacts the liquid water path as part of a cloud adjustment process, which then further alters the probability of precipitation. The relationship between aerosol-cloud interactions and cloud liquid water are neither universal nor well known. In order to reduce the uncertainty interpreting our results, we limit our observations to clouds with liquid water paths in a narrow range between 150 to 200 g/m$^{-2}$, building on work by Douglas and L'Ecuyer (2019) and Douglas and L'Ecuyer (2020) which found this LWP range to be an inflection point for cloud lifetime effects. In doing so, we focus only on how

aerosol alters the organization of rain formation and evaporation within the cloud layer, not its influence on cloud liquid water. Invigoration in this context includes how aerosol alters rain formation within the cloud, alters evaporation in the entrainment zones, and induces more turbulence.

Using latent heating and vertical motion profiles from the Wisconsin Algorithm for Latent heating and Rainfall Using Satellites (WALRUS), we show that there is a discernible signal of invigoration in warm clouds due to aerosol. Observations are

limited to cumulus clouds discerned using CloudSat and Cloud-Aerosol Lidar and Infrared Pathfinder Satellite Observation (CALIPSO) observations. The Moderate Resolution Imaging Spectroradiometer (MODIS) aerosol index (product of aerosol optical depth and Angstrom exponent) is used as a proxy for how aerosol concentrations affect the number of cloud condensation nuclei. A series of constraints are implemented in order to control for the role of stability in modulating (or confounding signals of) invigoration.

## 2   Data and Methods

### 2.1   Data

All observations are from instruments aboard NASA A-Train satellites from 2007 to 2010 and from 60° south to 60° north. Aerosol index (AI) from MODIS serves as our aerosol concentration proxy while the AMSR-E provides the mean cloud liquid water path of the scene. CloudSat's cloud profiling radar (CPR) is used to define cloud extent and we employ WALRUS to

infer changes in latent heating and vertical motion within cloud profile. CloudSat is limited by its temporal resolution, seeing the entire globe once every ~16 days compared to other Earth observing instruments like MODIS aboard Aqua which has a

daily resolution. By using multiple years of data from CloudSat, we can in some ways bypass the reduced temporal resolution, however it is possible that some rare phenomena will be missed by CloudSat or not well represented by our dataset.

Aerosol index is the product of the Angstrom exponent and the aerosol optical depth measured at 550 nm and is better correlated with cloud droplet concentrations than AOD (Ångström, 1964; Hasekamp et al., 2019). MODIS AI is available in clear sky scenes over the ocean, meaning cloudy AI must be interpolated from nearby cloud-free scenes (Levy et al., 2010). We remove AI within 2 km of the clouds in order to reduce the influence of aerosol swelling in high humidity scenes (Christensen et al., 2017). We define pristine conditions as those with an AI less than .042 and polluted as those with an AI higher than .09. These roughly correspond to the lower and upper 20 percentiles of our dataset. Avoiding intermediate AIs reduces the possibility our analysis captures possible transition states as clouds move out of the aerosol limited regime (Koren et al., 2014).

Clouds are limited to LWPs between 150 to 200 $\text{gm}^{-2}$ using AMSR-E (Wentz and Meissner, 2007). Although AMSR-E LWP is derived using a larger field-of-view, a rough constraint on cloud liquid water content provided the spatial extent and depth of the cloud are limited using CPR observations. Although we constrain LWP to homogenize the clouds observed, using LWP as a constraint introduces an uncertainty due to the effects of aerosol on LWP. There remains large uncertainties on how aerosol may increase (or decrease) LWP due to environmental confounders; these ignored effects may have led to changes in the eventual, precipitating cloud state (Gryspeerdt et al., 2019). Therefore, some uncertainty remains within our results as we do not control for this lifetime effect on invigoration. Cloud extents are defined using CloudSat's 2B-CLDCLASS-LIDAR product by sorting clouds by the number of contiguous cloudy pixels and limiting the analysis to clouds with at most 15 contiguous, cloudy pixels, approximately the size of an AMSR-E footprints (Sassen et al., 2008). This cloud based partitioning is analogous to the cloud object based partitioning used by Igel et al. (2014) except while Igel et al. (2014) focused on deep convective systems, our clouds are constrained to shallow convective types. We focus on cumulus warm clouds, rather than stratus or stratocumlus, in order isolate the effects of aerosol on shallow convection.

Environmental information is provided by MERRA-2 reanalysis. We define the stability of the atmosphere using the estimated inversion strength (EIS) (Wood and Bretherton, 2006).

$$EIS = LTS - \Gamma_m^{850} \times z_{700} + \Gamma_m^{LCL} \times LCL \tag{1}$$

where $\Gamma_m$ is the moist-adiabatic potential temperature gradient and LTS is the lower tropospheric stability.

Stability of the boundary layer controls the depth of the cloud making it imperative that this relationship is constrained in order to separate aerosol effects from environmental forcings (Zuidema et al., 2009). Unstable environments are defined as having an EIS below 1 K while stable environments are defined as having an EIS above 3 K. This partitions environments into two main regimes: trade cumuli (unstable) and cumuli from stratocumulus to cumulus transitions (stable). A dry free atmosphere alters the distribution of liquid throughout the cloud layer, thereby directly impacting precipitation formation processes as well. In order to control for these interactions, clouds are further subset into a dry regime whereby the $RH_{700}$ is below 30% to analyze how dry air entrainment may impact invigoration processes.

## 2.2 Latent heating profiles

The Wisconsin Algorithm for Latent heating and Rainfall Using Satellites (WALRUS) provides information on the latent heating and vertical motion profiles in the atmosphere. The algorithm combines CloudSat's CPR observations with a database of warm rain states derived from the Regional Atmospheric Modeling System (RAMS) simulations to emulate realistic latent heating rates and related vertical motion (Nelson et al., 2016). WALRUS limits our analysis to maritime clouds with heights less than the freezing level and only those that exhibit reflectivity greater than 0 dBZ somewhere in the column, consistent with the Rain Certain flag in CloudSat's 2C-PRECIP-COLUMN product (Haynes et al., 2009). Our results do not include the effects of drizzle on possible invigoration processes. This should also focus our results on only the growing and mature stages of shallow convection. Signals of invigoration are derived based on changes in the latent heating within the cloud, defined by WALRUS as the difference between the condensation and evaporation rates. Precipitation formation rates correspond to the latent heat release within the cloud, while evaporation due to entrainment at the cloud top or vigra below the cloud are indicated by cooling from WALRUS. We refer to the evaporation at cloud top as due to entrainment, however WALRUS does not simulate entrainment rates, therefore we are inferring from the evaporation at the top of the cloud profiles that this cooling is due to entrainment. Enhanced turbulence, or the change in vertical velocity, is determined by the difference in vertical velocity between polluted and pristine environments.

WALRUS is limited only to warm cloud precipitation, reducing our ability to understand mixed-phase convection. It is possible some of the rain events observed are the remnants of mixed-phase precipitation events that were unsuitable to infer latent heating profiles by WALRUS. Our conclusions drawn within are only for warm phase rain events. The latent heating profiles from WALRUS are based on a limited range of simulations from RAMS, meaning it is possible that some environmental states were not represented by the RAMS runs/WALRUS inference and could lead to some amount of error in our analysis. We limit our observations to only rain certain scenes, discarding drizzling and higher rain rate observations that may attenuate the CloudSat signal. This reduces some of the uncertainty due to a model derived, observationally based product. As Nelson and L'Ecuyer (2018) have also commented, the results herein could instead be reframed as how the RAMS microphysics scheme would map onto real observations of global precipitation.

WALRUS employs a Bayesian Monte Carlo method in order to derive probabilistic latent heating profile. While precipitation amounts alone can be used to infer total latent heating in the column, vertically-resolved reflectivity profiles allow the inference of the distribution of latent heating throughout the profile, below, within, and above the cloud. The Bayesian Monte Carlo method relies on an a priori distribution of possible characteristics to connect to the CloudSat observations. The a priori database is created using the RAMS model with simulations based on the Atlantic Trade Wind Experiment field campaign. The model is run at a 250 m horizontal and 100 m vertical resolution for a set of sea surface temperatures (293 K, 298 K, and 303 K). Quick Beam produces radar reflectivity profiles and attenuation signals from the RAMS simulation, which are sampled every 40 minutes for the database. Overall, WALRUS had 1.4 million possible a priori warm rain structures against which observed CloudSat reflectivities are compared to retrieved the most physically realistic latent heating and associated vertical motion rates. For more information please refer to Nelson et al. (2016).

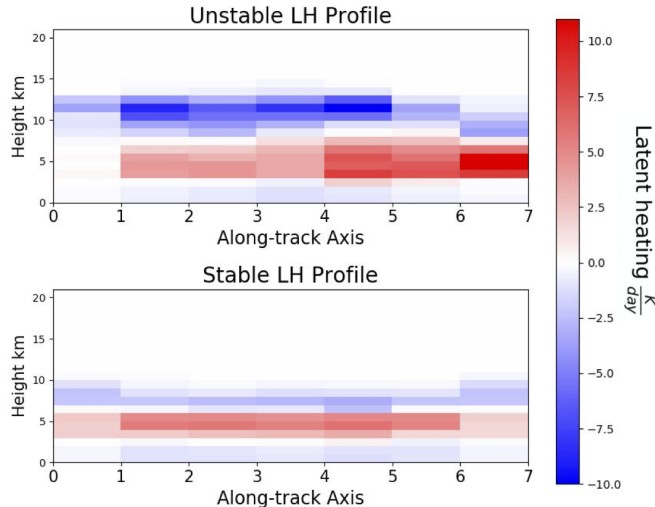

**Figure 1.** An example of the latent heating profiles for clouds in unstable (top) and stable (bottom) environments with a 7 km rain system size from warm cloud systems over the Pacific in January 2007.

## 2.3 Partitioning clouds

Cloud profiles are partitioned according to the individual cloud base and cloud top heights determined for each profile using
CloudSat's 2B-CLDCLASS-LIDAR CloudLayerBase and CloudLayerTop products. These heights are used to distinguish the
in-cloud region from the environment below or above it. The maximum above cloud cooling due to evaporation is found by
taking the maximum of all evaporative cooling rates starting at the cloud top to the top of the profile. The cloud top is obvious
in the latent heating profiles (Figure 1); the abrupt shift from heating to cooling indicates the entrainment zone of the cloud near
the cloud top. The mean below cloud cooling rate is similarly found using the cloud base height from 2B-CLDLCASS-LIDAR
and taking the sum of all evaporative cooling rates at the cloud base to the bottom of the profiles (approximately ground level).
The geometrical center of the cloud is the midpoint of the cloud (e.g. for a 7 km cloud as seen in Figure 1, the midpoint is 3.5
km) therefore the profiles on either side are used to determine the behavior of the geometrical center of the cloud.

## 3 Results and Discussion

### 3.1 Aerosol effects on warm rain formation rates

Theoretical arguments for warm rain invigoration predict that in a more polluted environment, the rate of collision coalescence
and therefore precipitation production increases. Our analysis suggests that, on average, clouds in polluted environments do
not show an increased rates of precipitation formation relative to those in pristine environments (black solid line, Figure 2). The
difference between polluted (solid) and pristine (dashed) conditions is minimal when clouds in environments are considered

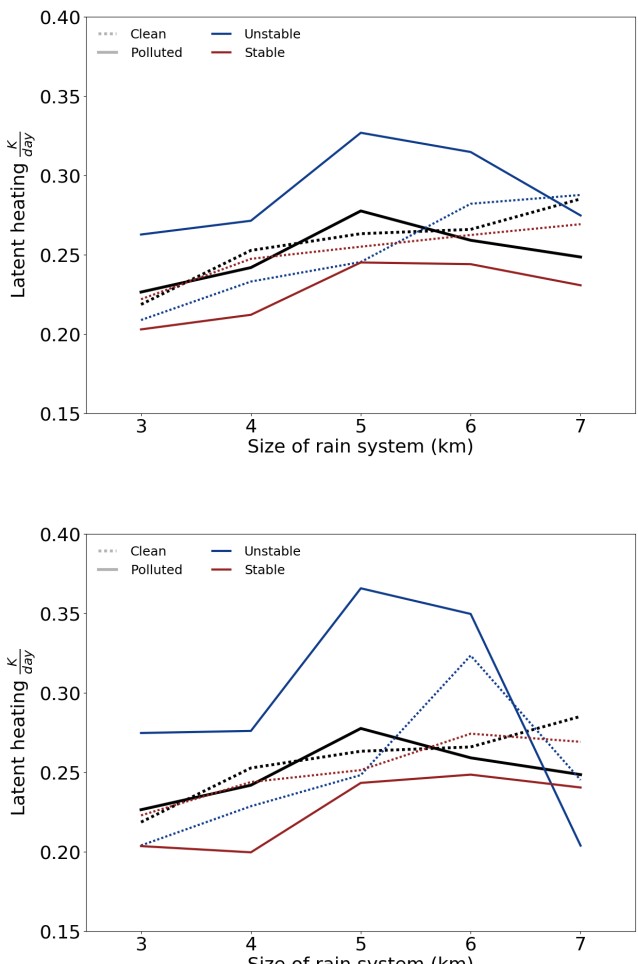

**Figure 2.** The mean amount of latent heating released due to precipitation formation rate in the geometrical center of the rain system as a function of rain system size for all (top) and dry (bottom) warm clouds with an extent of 15 km. Black is for all stabilities, blue is for unstable environments, red is for stable environments; dashed represents pristine and solid represents polluted surroundings.

together. However, when separated according to the environmental stability, it is evident that the reason for this is not that the warm rate intensity is unaffected by aerosol loading, it is that clouds react differently under stable and unstable conditions. In unstable environments, polluted conditions lead to a marked increase in precipitation rate relative to unstable, pristine conditions (blue, dotted line) for all rain systems smaller than ∼6 km. Conversely, stable, polluted conditions (red, solid line) lead to a decrease in precipitation rate relative to stable, pristine conditions (red, solid line). The opposite reactions in stable vs. unstable conditions offset each other, giving the impression that warm rain is unaffected by aerosol loading when in actuality its sensitivity is environmentally dependent. Invigoration is only identifiable when stability is accounted for and this suggests that aerosol-induced invigoration of shallow convection may exhibit marked spatial patterns globally. While our analysis does account for some amount of covariation between meteorology and aerosol-cloud interactions, there is some added uncertainty due to the inherent relationships between aerosol and meteorology, as certain meteorological conditions may lead to high or low aerosol loading.

Polluted clouds exhibit first an increasing, then decreasing relationship between the size of the rain system and the maximum latent heating within the cloud; while pristine clouds show a steady, linear increase in the rain formation rate as the size of the rain system increases (Figure 2). Rain formation in polluted clouds appears to increase with rain size up to 5 km, then either decrease (unstable, dry) or remain constant with size. This inflection point, whereby the behavior changes from increasing with rain size to decreasing, depends on both the stability of the boundary layer and the humidity of the free atmosphere. Pristine conditions do not show this same pattern, as for all meteorological conditions, an increase in rain size results in an increase in maximum heating.

The core of a warm convective system should theoretically exhibit the greatest invigoration of precipitation. Our results indicate this conceptual model is correct: as invigoration of the warm rain formation rate due to aerosol is most pronounced in the geometrical center of the rain system (Figure 3). Mean precipitation rates increases in the center of unstable, polluted clouds relative to both cleaner and more stable conditions. This effect is exacerbated in dry conditions (Figure 3, bottom) until the rain system seems to hit a size inflection point around 7 km. While instability in polluted clouds leads to greater formation rates in the center, clouds in stable but equally polluted environments show a decrease in rain production relative to pristine conditions. This supressive behavior is observed regardless of the overlying free atmosphere, as clouds in dry environments (Figure 3 bottom) show the same behavior as all clouds (Figure 3 top).

## 3.2 Aerosol effects on evaporative processes

That is not to say that the free atmosphere does not play a role in altering the thermodynamics or possible invigorate state of warm rain systems. Evaporative processes link entrainment, below cloud evaporation, precipitation formation, and the energy budget of a cloud. When focusing on how aerosol may affect entrainment, the moisture content of the free atmosphere becomes a controlling factor. A drier atmosphere fosters greater evaporation rates above the cloud in more polluted environments (Figure 4). Cumuli generally have large rates of lateral entrainment that would not be captured by WALRUS, however lateral entrainment would also affect the invigoration of any rain formation in the cloud layer. While increased mixing with the free atmosphere may lead to cloud deepening, when the boundary layer unstable, it may also lead to an early onset of cloud breakup

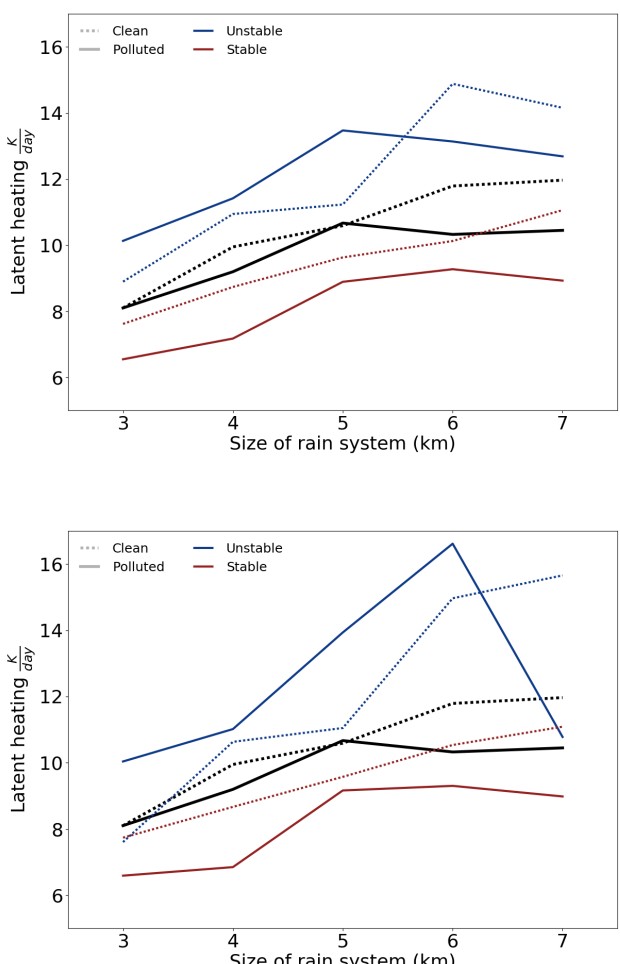

**Figure 3.** The maximum rate latent heat release due to precipitation within the cloud as a function of rain system size for all (top) and dry (bottom) warm clouds with an extent of 15 km. Black is for all stabilities, blue is for unstable environments, red is for stable environments; dashed represents pristine and solid represents polluted surroundings.

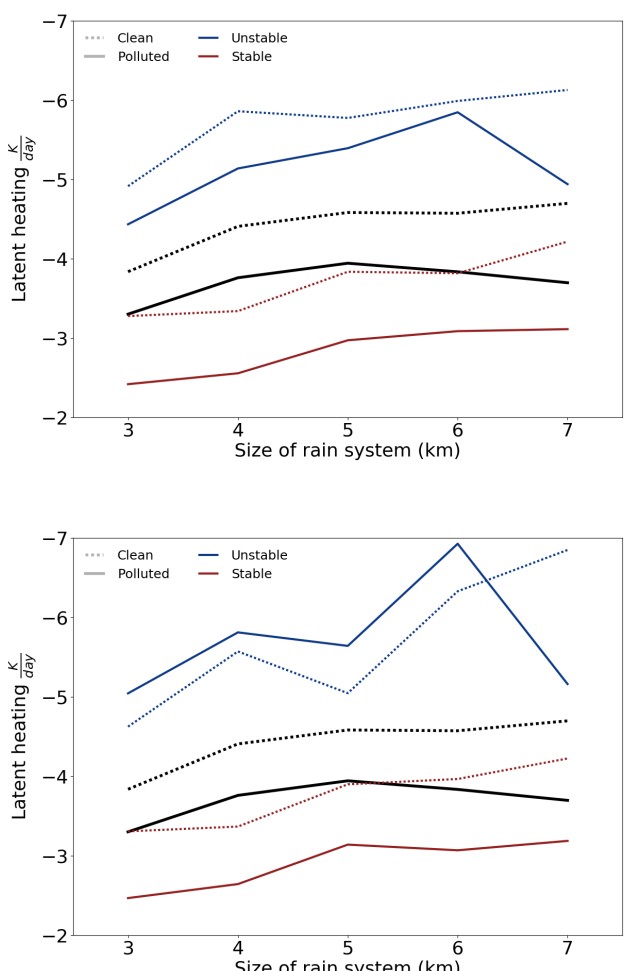

**Figure 4.** The maximum rate latent heating released due to evaporation above the cloud as a function of rain system size for all (top) and dry (bottom) warm clouds with an extent of 15 km. Black is for all stabilities, blue is for unstable environments, red is for stable environments; dashed represents pristine and solid represents polluted surroundings.

processes through evaporation-entrainment (Small et al., 2009). In some cases, increased entrainment and evaporation at the cloud top could lead to reduced cloud top heights, opposite of an invigoration effect (Xue and Feingold, 2006). Whether the growth of a particular cloud is enhanced or inhibited may depend on the distribution of liquid water near the cloud top and the ability of the cloud to penetrate the free atmosphere.

A drier atmosphere enhances cloud top evaporation in only unstable conditions; clouds in stable conditions are unaffected by a drier free atmosphere. This is likely due to the stronger capping inversion in stable conditions which limits mixing with the dry free atmosphere, limiting its effects on the cloud layer and, by extension, the invigoration process (Christensen and Stephens, 2011). While clouds in stable environments have similar responses in precipitation formation rates, inversion strength acts to limit evaporation at the cloud top. In theory, by limiting the amount of mixing with the free atmosphere, the inversion damps the ability of stable, polluted clouds to deepen compared to unstable, polluted clouds. While these clouds do not show signs of invigoration, stable, polluted conditions may prolong cloud lifetime by lessening cloud thinning processes (Van der Dussen et al., 2014).

That the cores of dry, polluted, unstable systems experience significantly greater rain formation rates than all other environments may suggest these clouds undergo some aggregation process focusing the majority of precipitation formation within the core of the cloud. This results agrees with a theoretical model posed by Morrison (2017), where entrainment of dry air leads to narrowing effect on the cumuli and enhancement of the core. Aerosol may act to invigorate this specific response by increasing the entrainment-evaporation at the cloud top, promoting turbulence within the cloud layer.

Below cloud evaporation and its associated cooling destabilizes the boundary layer, which could then further invigorate the cloud layer through amplified turbulence (Xue and Feingold, 2006). The effects of below cloud evaporation on the stability are sensitive to the location of the cooling and drop size of the cloud; it is possible in some circumstances that cooling can act to stabilize the boundary layer. Figure 5 demonstrates that larger warm rain systems exhibit considerably more below cloud evaporation than smaller systems. Evaporation, even when weighted by the size of the rain system, scales with the total amount of rain forming, as more rain means more possible below cloud evaporation. Polluted clouds exhibit less below cloud evaporation regardless of the stability and size of the rain system. This may imply that pristine conditions stabilize the boundary through below cloud cooling and increasing the boundary layer temperature inversion.

There are two possible mechanisms that may lead to polluted environments having a relatively lower rate of evaporation below cloud compared to their pristine counterparts. The first mechanism relies on the change in droplet size due to the differences in where rain is being produced in the cloud under pristine and polluted conditions. In polluted conditions, rain may form higher within the cloud; as precipitation forms and drops fall, the drop grows larger as its path increases, decreasing the amount of evaporation below cloud base (Dagan et al., 2016). So although aerosol loading decreases the mean cloud drop size, rain droplet size experiences an inverse effect, the magnitude of which is determined by the height within the cloud where precipitation forms.

The second possibility is that in environments with drier free atmospheres, clouds in unstable environments (both pristine and polluted) have much greater rates of evaporation below the cloud (Figure 5 top). The increased evaporation below cloud may be driven by vertical motion forced by increased evaporation in the entraining layer of the cloud (Figure 4), leading to

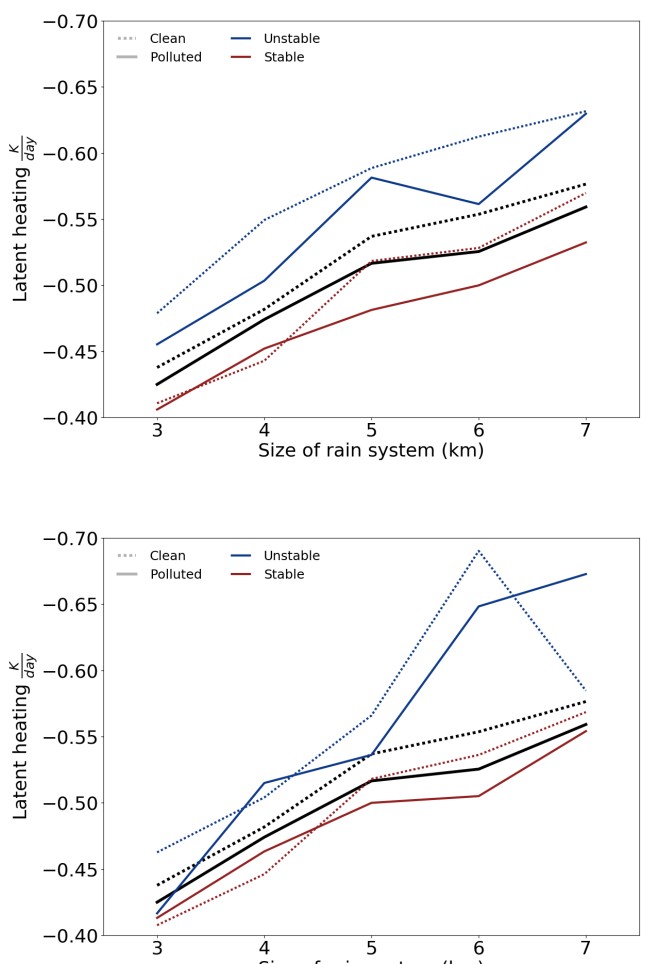

**Figure 5.** The mean rate of latent heating due to evaporation below the cloud as a function of rain system size for all (top) and dry (bottom) warm clouds with an extent of 15 km. Black is for all stabilities, blue is for unstable environments, red is for stable environments; dashed represents pristine and solid represents polluted surroundings.

more mixing throughout the cloud layer. This in turn leads to more activation of available CCN, decreasing the mean drop size and increasing the rate of evaporation below the cloud in both polluted and pristine cases. It is possible that in polluted, unstable environments, smaller droplets evaporate efficiently, quickly increasing the humidity of the lower boundary layer, resulting in an overall decrease in the rate of evaporation (relative to pristine conditions) as the cloud continues to precipitate (Pincus and Baker, 1994; Jiang et al., 2009). Or it may be that stable, polluted boundary layers show the lowest rates of below cloud evaporation because they also have lower rates of precipitation formation. Establishing the specific processes responsible for the observed invigoration signatures is not possible from current satellite observations.

The implied effects on cloud lifetime in unstable, polluted conditions agree with Albrecht's original hypothesis, whereby aerosol works to increase cloud lifetime. The overall effect of aerosol loading on not just the thermodynamics, but lifetime of the cloud depend heavily on the environment surrounding the cloud (Albrecht, 1993). In specific high polluted conditions, whereby the boundary layer begins unstable, it could be possible that below cloud evaporation cools the lower boundary layer, while latent heating due to rain formation warms cloudy portion of the boundary layer, which would act to stabilize an unstable the boundary layer. This stabilizing effect would decrease in time, however, as the magnitudes of both the cooling below cloud and warming in cloud depend on the instability (Dagan et al., 2017). The same stabilizing effect may be seen in pristine scenes as well, as clean clouds in unstable conditions also showed greater rates of below cloud cooling due to evaporation. So while aerosol may work to help prolong lifetime through this stabilizing mechanism, the environment through a stabilizing feedback works to lengthen the cloud lifetime regardless of the aerosol conditions.

## 3.3 Aerosol Effects on Vertical Motion

Results are consistent with the hypothesis that invigoration will increase vertical motion, which may lead to an increase in turbulence, as indicated by changes in vertical motion, within the cloud layer due to greater amounts of latent heat release (Rosenfeld et al., 2008). Figure 6 shows that clouds in polluted environments display higher updraft speeds within the cloud layer than those in pristine environments. This reaffirms an ongoing hypothesis that cloud deepening is driven by enhanced updrafts (Christensen and Stephens, 2011). Aerosol may act to redistribute water throughout the cloud resulting in changes to the distribution of latent heating (Dagan et al., 2018). Modifying where latent heat is released, especially increasing the difference between the center where rain formation is occurring and edge evaporation due to cloud edge entrainment, alters vertical motion and flow within the cloud layer. As seen in Figure 1, edge and core behavior and latent heating signatures are markedly different. While it remains unclear how aerosol may affect the absolute amount of water within a cloud, it is clear aerosol affects how water is distributed within the cloud (Toll et al., 2019; Rosenfeld et al., 2019).

When separated into stable and unstable environments (Figure 6) it becomes obvious how *strongly* stable environments damp invigoration. While unstable environments intensify the vertical motion within the cloud layer, stable environments show only a faint increase in vertical motion in the center of the cloud. This may explain why stable, polluted environments also manifest the smallest rates of evaporation due to cloud top entrainment, as these clouds have less overturning motion near the cloud tops. Though stable, polluted environments display a reduced core precipitation formation rate compared to their pristine counterparts, the reduced size of the polluted droplets may allow greater mobility and therefore vertical motion (Koren et al.,

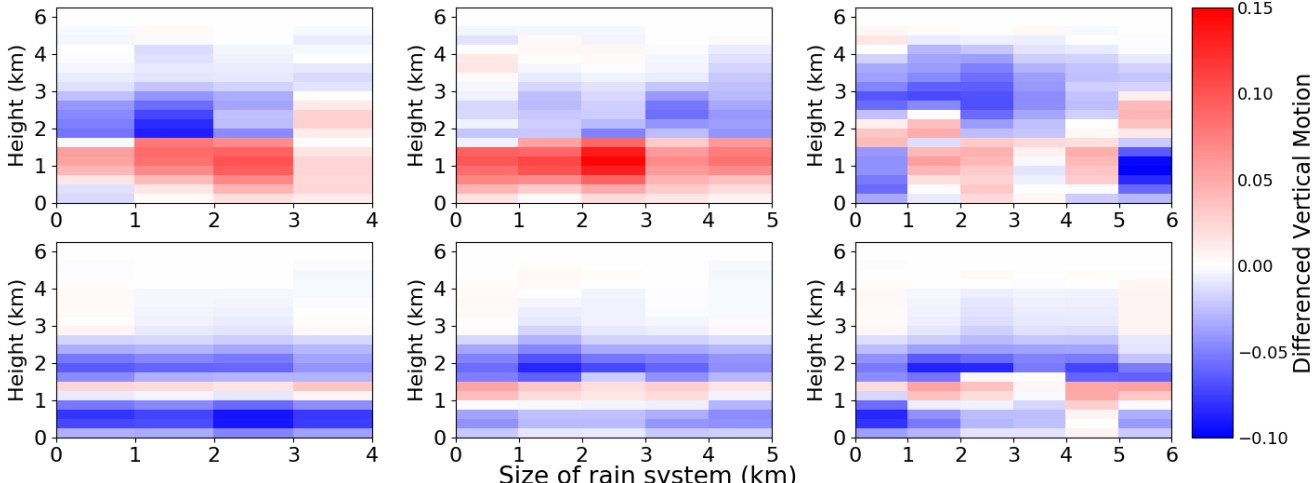

**Figure 6.** The difference (polluted - pristine) in vertical motion in unstable (top) and stable (bottom) environments for clouds of rain size 4 km (top), 5 km (middle), and 6 km (bottom).

2015). Unstable, large rain systems (∼ 6 km) may have downdrafts within the cloud due to enhanced evaporation entrainment, leading to more in-cloud turbulence, overturning motion, and mixing.

It should be noted that increased rates of rain formation in unstable conditions could induce a positive feedback: latent heating increases from faster rain formation which leads to more vertical motion and turbulence, greater updraft speeds, more entrainment and therefore buoyancy fluctuations within the cloud, which in turn leads to more collision-coalescence and latent heating. This could explain both the greatly enhanced rain formation rates in the center (Figure 2) and intensified turbulence (Figure 6). However, this feedback may be particularly sensitive to the size of the rain system, as larger systems may decrease the chance of the feedback occurring. The scattered increases in turbulence seen in rain systems of size 6 km and the sloping off of rain formation rates at the same size imply a sensitivity to the size and organization of the system (Fan et al., 2016). It is possible this point may also represent how different clouds experience different "optimal" conditions for invigoration, as these larger clouds within the same environment may have a higher AI that would be needed to experience the same invigoration as their small, environmental counterparts (Liu et al., 2019). Additionally, any possible invigoration feedback may be moderated by a stabilizing effect induced in unstable conditions, whereby increased cooling due to evaporation below cloud and warming due to rain formation within the cloudy layer stabilize the boundary layer.

## 4 Conclusions

Cumulus clouds in polluted, unstable environments display greater rates of maximum observed and average core precipitation formation along with greater amounts of vertical motion and therefore turbulence within the cloud. Dry environments act to

increase this response, along with inducing further invigoration effects by increasing the amount of entrainment mixing. Stable environments act to inhibit any invigoration by capping entrainment effects and reducing precipitation formation rates. In polluted environments, a stable boundary layer and strong inversion acts to inhibit rain production relative to pristine environments. This reverse response may be driven by reduced amounts of vertical motion in polluted cloud tops and bases, hindering precipitation formation throughout the cloud.

Invigoration is an "elusive" effect in the aerosol-cloud interaction community perhaps because observing it depends on the definition used. Based on our definition, whereby aerosol loading in warm clouds increases the precipitation formation rate and in-cloud vertical motion (a proxy for turbulence), there is evidence invigoration may occur. However from the results shown, two important aspects of invigoration emerge:

1. The magnitude of invigoration in marine cumuli depends strongly on the size of the rain system. As the size of the rain system increases, all possible signs of invigoration, from more rain formation to increased turbulence, are a function of the size of the rain system. This implies the organization of the cloud plays a role in defining how aerosol loading may impact aspects of invigoration. The dependence of these processes on the size of the system may explain why many components of invigoration (LWP response, cloud deepening, etc.) are non-linear when regressed against aerosol alone.

2. Stability can reduce and/or reverse all aspects of invigoration within the cloud layer. The mean warm cloud signal of invigoration is completely buffered by the environment unless stability is accounted for.

The mean formation rate in polluted environments closely tracked the mean pristine rate (Figures 2, 3), and it is only by accounting for both the effects of stability and humidity do distinct signals appear. Even with stability and the environment accounted for through sets of constraints, we had to impose limits on liquid water path in order to isolate the invigoration. Without these constraints, on the environment and cloud state, invigoration would be indiscernible using only the mean response. Models must account for these factors when parameterizing aerosol impacts on precipitation.

Our analysis is heavily predicated on latent heating and vertical motion from WALRUS. As such, some uncertainty is inherent in the results due to the indirect nature of a satellite-based latent heating and vertical motion estimates. However, the vertical structure of reflectivity and integral constraint provided by the path-integrated attenuation (PIA) provide strong constraints on hydrometeor distributions and integrated water path in warm clouds (Lebsock and L'Ecuyer, 2011; Nelson et al., 2016). The results presented here imply that aerosols induce systematic changes in observed reflectivity profiles and attenuation from raindrops that are indicative of different precipitation formation and above/below cloud evaporation rates, though the precise magnitudes may be uncertain. Further support for the plausibility of the WALRUS products is provided by Nelson and L'Ecuyer (2018) who document systematic regional variations in latent heating and inferred vertical motion in global warm rain systems. This study is limited to only a "snapshot" view of clouds, unable to account for the individual lifetimes of individual cumuli. Tracking clouds throughout their lifetime, similar to the tracking employed by (Christensen et al., 2020) to follow clouds through the stratocumulus to cumulus transition, would offer insights into regime specific processes at the root of the signatures seen here. However, geostationary satellite observations provide much more limited insights into cloud

structure. The lack of vertical structure precludes the retrieval of in-cloud latent heating and vertical motion. Future analysis should attempt to blend vertical structure information with observations of the state of the cloud throughout its lifetime in order to understand how the trajectory of a cloud affects its response to aerosol loading. Additionally, we aimed to reduce uncertainty by only contrasting high and low aerosol loading scenes. An aim of future work should include defining the patterns of changes in relation to incremental increases in aerosol to better define and understand these relationships.

*Author contributions.* Alyson Douglas completed all analysis and writing of the paper. Tristan L'Ecuyer provided guidance on the analysis and critical feedback on the story of the paper.

*Competing interests.* The authors have no competing interests to declare.

*Acknowledgements.* Thank you to Guy Dagan for feedback on the manuscript. Thank you to Philip Stier and the EU Horizon Grant for funding of this work. Thank you to the NASA CloudSat Science Team for initial funding of this work.

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

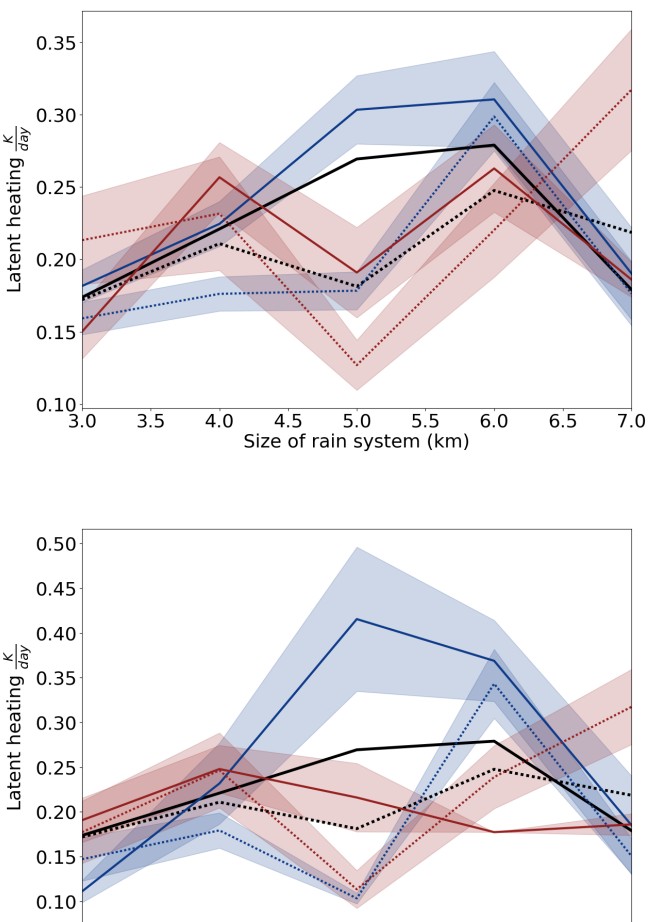

**Figure A1.** Figure 2 with ± standard error shaded.

Zuidema, P., Painemal, D., De Szoeke, S., and Fairall, C.: Stratocumulus cloud-top height estimates and their climatic implications, Journal
of Climate, 22, 4652–4666, 2009.

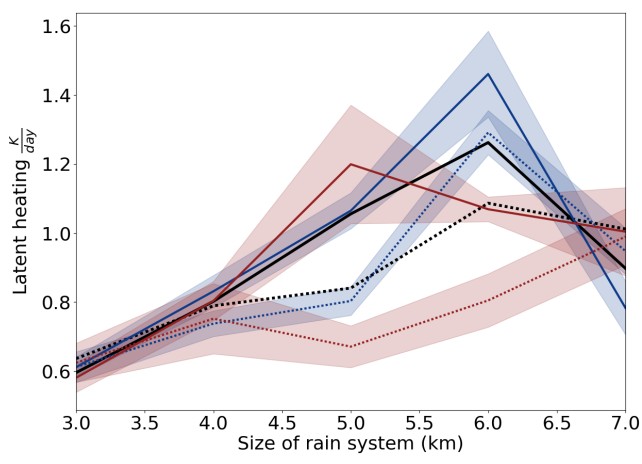

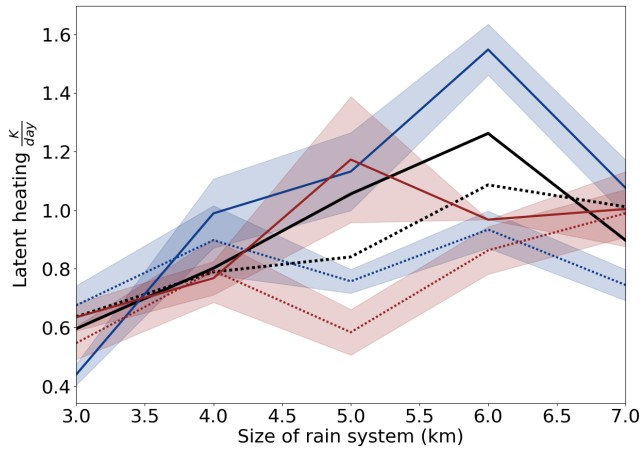

**Figure A2.** Figure 3 with $\pm$ standard error shaded.

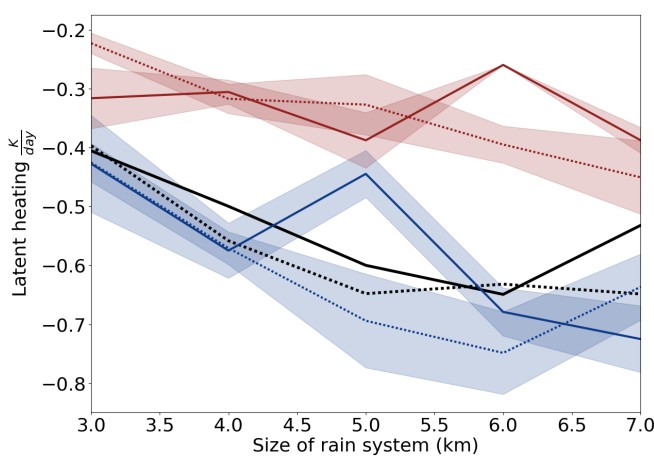

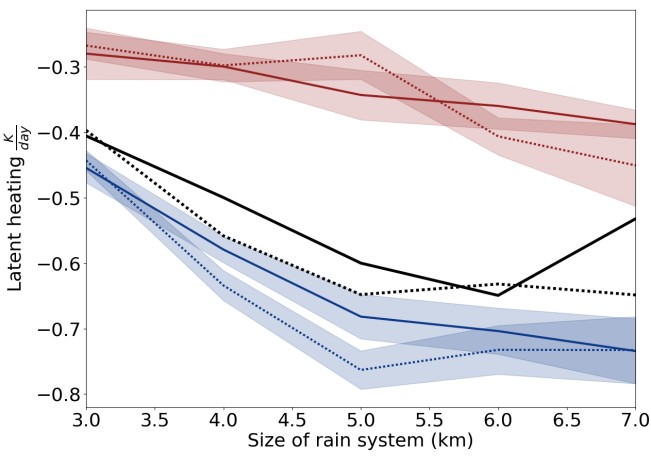

**Figure A3.** Figure 4 with ± standard error shaded.

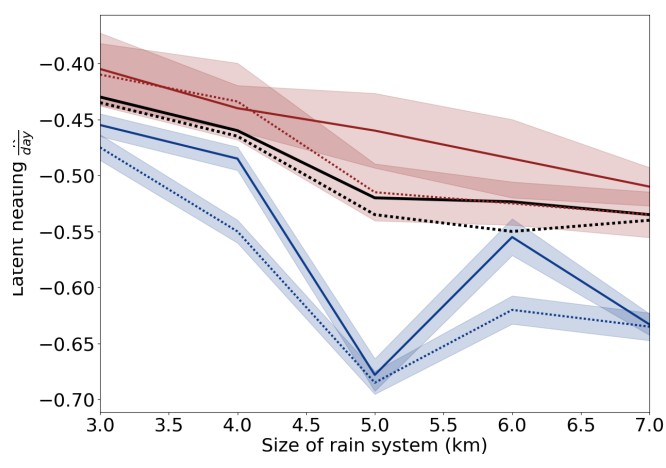

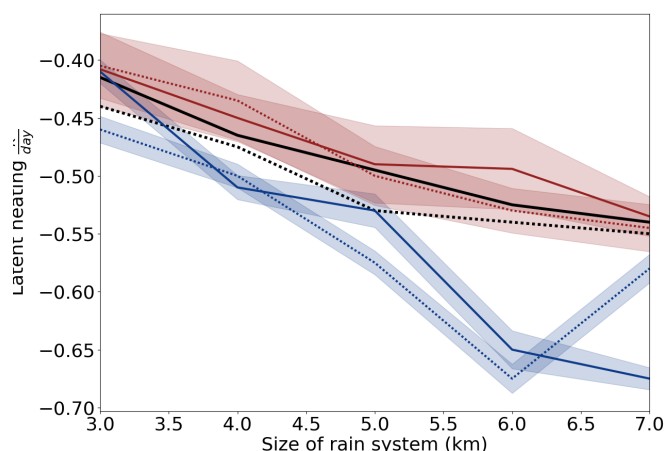

**Figure A4.** Figure 5 with $\pm$ one standard error shaded.