# Peer review of "Global Evidence of Aerosol-induced Invigoration in Marine Cumulus Cloud"

_Atmospheric Chemistry and Physics, 2021_

## Author Comment (AC1)

**Subdividing the AI into two end-groups:** Such separation may miss cloud sensitivity to the optimal aerosol loading. As the authors know (and cite) aerosol effects on clouds were shown to be non-monotonic. A competition between core (invigorating) and periphery (enhance evaporation and entrainment) processes dictates an "optimal aerosol concentration". It means that properties like maximal cloud depth, updrafts, maximal LWP, and rain may have their maximal values in a given aerosol concentration (the "optimal" concertation (or optimal AI) of the invigorated branch) and a further increase in the concentration will enhance evaporation and entrainment and therefore will result in lower values of these key properties. Previous work has suggested that the optimal AI depends on environmental conditions. Clouds in a more unstable environment will "enjoy" higher optimal AI. Theoretically, it is possible that the optimal AI is missed in the central values that are not analyzed here and the trends shown in this work are samples of the beginning of the ascending branch vs. the end of the descending. The sensitivity to the environmental properties could be much richer than what is shown here.

We agree that there is a sensitivity that non-linearly increases with aerosol loading. The slope, and the points where this relationship goes from linear with log(AI) to non-linear with log(AI) are dependent upon a number of factors, including some of the same environmental constraints we use within (RH700 and EIS). We aimed to show that there are marked differences between the most polluted warm rain events and the least polluted warm rain events, even after holding the environment that modulates both the response to aerosol and the formation of rain approximately constant. Further steps, as mentioned in our conclusions, would be to follow clouds throughout their lifetime to identify the environment and the aerosol conditions that lead to each, unique response. We consider this work a first step in understanding how the environment leads to different responses and what the approximate differences would be between clouds that form/precipitate/dissipate in a polluted vs. clean environment.

We have added in the conclusions, when discussing the sloping off of the effect in section 3.3:
" It is possible this point may also represent how different clouds experience different "optimal" conditions for invigoration, as these larger clouds within the same environment may have a higher AI that would be needed to experience the same invigoration as their small, environmental counterparts (Liu et al. 2019)."

**LWP slicing:** While holding one key variable and checking sensitivity in other variable approach makes sense when looking at the aerosol concentration (or the AI) as a measure for CCN as a continuous variable, here the subdivision to two end groups may infer problems. Along the same lines as the first comment, changes in aerosol concentration can strongly affect LWP. Therefore by comparing clouds in two very different aerosol concentration regimes with a similar LWP, may imply that under the same aerosol conditions these clouds will have a completely different LWP and therefore evolve in different thermodynamical conditions. This should be noted and considered in the analysis.

We agree that it should be noted within the manuscript that LWP is affected by aerosol loading, and by using it to subset our observational dataset we may have introduced a bias that doesn not account for the effects of aerosol on LWP.

We have added to the methods when first mentioning the use of LWP constraints in section 2.1:

"Although we constrain LWP to homogenize the clouds observed, using LWP as a constraint introduces an uncertainty due to the effects of aerosol on LWP. There remains large uncertainties on how aerosol may increase (or decrease) LWP due to environmental confounders; these ignored effects may have led to changes in the eventual, precipitating cloud state (Gryspeerdt et al. 2019). Therefore, some uncertainty remains within our results as we do not control for this lifetime effect on invigoration."

The study area (60S to 60N) is large. It includes the tropics, subtropics, and mid-latitudes. Marine warm clouds over this region can be very diverse. They depend on the SST, the MBL properties, winds, free atmosphere properties, as well as aerosols. Trends in cloud properties can be related also to the cloud location – wouldn't it be better to limit the study area to the subtropic? Or at least the authors should show that there are no correlations between the cloud properties and their geographical area.

The study area is large, however most of our observations reside in the trade cumuli regions of the southeast Pacific, south Atlantic, and near Hawaii. The constraints we have on our observations (warm cloud, size 15 km, between 150 to 200 gm-2) inherently specifies regions where these clouds occur. In future work, instead of setting hard limits in order to specify the cloud type chosen, we may instead work to use regimes of cloud controlling factors (https://ui.adsabs.harvard.edu/abs/2021EGUGA..2316443D/abstract).

On a similar note to the comments above, classifying the environment to only two states (stable and unstable) using EIS threshold is a bit limited. Theoretically, there are many types of profiles that can be regarded as stable and many others as unstable and these profiles may yield different clouds. Limiting the study area to a more uniform one (with respect to environmental conditions) can solve this issue.

During our analysis, we aimed to show the most extreme behavior in order to prove that there are fundamental differences in latent heating profiles even when meteorology is controlled for. In a more broad study focusing only on the effects of stability on precipitation or on only the effects of RH in FA on precipitation, we could go into how each environmental parameter works to affect aerosol-cloud-precipitation interactiosn. Nelson et al showed how stability affects the different cloud profiles of precipitation in global warm clouds without limits on vertical extent. Our limits of vertical extent of the precipitation is a strong control that homogenizes the stable vs. unstable profiles more than only separating stable from unstable. It is the use of multiple constraints that lets us compare these profiles. But we agree, that there is variation seen within different ranges of stability, and have added to our methods when mentioning the constraints:

There exist a large range of effects depending on the stability. Separating only by stable or unstable may lead to some error due to the range of effects that could be seen within each

regime. However, by combining constraints (on LWP, RH, and rain size) we can somewhat account for the range of effects seen within a single regime of EIS, RH, LWP, or rain size.

Many space instruments are being used in this analysis – I miss a critical discussion on their limitations. I miss a discussion on possible biases due to measurement limitations.
Many of the products used throughout the analysis are commonly used by the observational community to discern similar aerosol-cloud interactions. It is out of the scope of this paper to do a full analysis of each instruments' limitations and biases. We have added caveats that address the main limitations of the two main sources of observations within:
-When discussing CloudSat in section 2.1:
"CloudSat is limited by its temporal resolution, seeing the entire globe once every ~16 days compared to other Earth observing instruments like MODIS aboard Aqua which has a daily resolution. By using multiple years of data from CloudSat, we can in some ways bypass the reduced temporal resolution, however it is possible that some rare phenomena will be missed by CloudSat or not well represented by our dataset."

When discussing WALRUS in section 2.2:
"WALRUS is limited only to warm cloud precipitation, reducing our ability to understand mixed-phase convection. It is possible some of the rain events used are the remnants of mixed-phase precipitation events that are unsuitable to infer latent heating profiles by WALRUS. Our conclusions drawn within are only for warm phase rain events."

Estimate inversion strength (EIS) is a key variable – would be nice (and not too complicated) to provide details on exactly how it is calculated.

We have added the formula from Wood and Bretherton 2006 that discusses how to calculate EIS to section 2.1 Data. The properties needed to calculate EIS are provided by MERRA-2 as mentioned within the text.

---

## Author Comment (AC2)

Given the key roles of the non-conventional products of latent heat and updraft, it is key to demonstrate that the uncertainties of these products won't outweigh the signals of the aerosol invigoration. Referring to the paper introducing these products (WALRUS) (Nelson et al., 2016), their uncertainties seem so large whose impact on the findings of this study may be overwhelming.

We agree that the uncertainty in certain derived products can be large. We have added an appendix with shaded ranges of error. To quote Nelson & L'Ecuyer 2018,

Assumed errors are consistent with observation uncertainties and algorithm resolution: reflectivities are 1 dBZ, attenuations are 2 dB, and heights are 300 m. Since the reflectivity structure provides strong constraints on both the vertical structure of hydrometeors and column-integrated water content, errors in model physics are not likely to exert a prohibitive influence on the retrieval. The algorithm simply requires that the RAMS database adequately span a range of atmospheric scenes that may be encountered in nature and lets the observations define the relative frequency. Nevertheless, it is important to note that the process rates analyzed below derive from a model database. At worst, WALRUS can be considered as providing a framework for mapping state-of-the-art RAMS microphysics globally.

Further, the uncertainties from WALRUS in some cases may be large when the signal is attenuated or the cloud structure is not well resolved. We do not include cases with high rain rates or drizzle in order to reduce the uncertainty in the WALRUS derived quantities. We have added to section 2.2

"We limit our observations to only rain certain scenes, discarding drizzling and higher rain rate observations that may attenuate the CloudSat signal. This reduces some of the uncertainty due to a model derived, observationally based product. As Nelson & L'Ecuyer have also commented, the results herein could instead be reframed as how the RAMS microphysics scheme would map onto real observations of global precipitation."

In all the figures, only the mean curves are drawn without any measures of variations. I'd suggest to present some scatter plots showing the real distributions of the data points, and add standard deviations for all the rest, together with significance tests to see if the differences among the curves are significant at certain level of confidence. The colors of the different curves are too close to differentiate.

In order to clearly represent the differences, we chose to use mean curves. To show the variation in the plots, we have added versions of the figures to an appendix with shading to represent the standard error. We chose the colors to have the same intensity, which helps those who are differently abled to differentiate the lines. An example of how someone with a weak red/green colorblindness would see the figures is shown here.

If green or purple were introduced to color by aerosol loading as well, the differences between the two lines would become less obvious for some with some amount of colorblindness. Therefore, we chose to delineate using dashes and a small subset of colors.

Now that turbulence is estimated from updraft speed, it is somewhat misleading to state evidences are found for both updraft and turbulence. Of course, such inference itself is debatable that induce more uncertainties.

We did not intend to imply both turbulence and updraft were increasing, rather that because updraft speed is increasing, it is more likely turbulence is increasing. We have corrected all usage of the term turbulence within the paper, specifically in section 3.3 where vertical motion is discussed, to instead say vertical motion which may imply turbulence.

The inference of latent heat was based on the RAMS modeling data which are highly limited to a handful of cases at very few locations. How much error may be incurred for this global application study ?

There is some uncertainty due to the limited environmental states represented by the range of RAMS simulations, however we do not think these would significantly alter the results. Saleeby et al. 2015 showed that a range of RAMS simulations can capture a range of shallow convection processes and environmental states. Further, they verified that the RAMS runs agree well with others who have simulated various cases of shallow convection. We have added to our section on WALRUS (2.2)

"The latent heating profiles from WALRUS are based on a limited range of simulations from RAMS, meaning it is possible that some environmental states were not represented by the RAMS runs/WALRUS inference and could lead to some amount of error in our analysis."

All the findings are shown with respect to size of rain system which does have some merit to this study in disclosing the dependence of the invigoration on cloud size. It would also be valuable and revealing if some findings are given w.r.t. aerosol loadings as in many previous studies.

In order to reduce the uncertainty from the non-linear relationships between warm rain suppression processes and aerosol, we chose to only contrast high and low aerosol while taking into account some of the cloud organization by using the rain size as a constraint. Future work may explore these relationships in more detail while adding more information on organization, similar to how Janssens et al. 2021 showed four dimensions (including a statistic similar to ours based on cross track cloud size) can be used to explain multiple cloud features. This work is preliminary work to show that there are differences in the latent heat and vertical motion that may be explained by the differences in aerosol state, after constraining the environment and rain size. We have added to section 4, conclusions:

"Additionally, we aimed to reduce uncertainty by only contrasting high and low aerosol loading scenes. An aim of future work should include defining the patterns of changes in relation to incremental increases in aerosol to better define and understand these relationships."

---

## Author Comment (AC4)

It is important to state here that latent heating in this study refers to the net latent heating of the system, which is not the same as the latent heating in an updraft. Weakly precipitating marine cumulus and stratocumulus have considerable latent heating in updrafts and cooling in downdrafts, and yet have close to zero net latent heating. These can be quite vigorously overturning layers. **Precipitation suppression by aerosol in these clouds can drive stronger turbulence (see e.g., Ackerman et al., 2004) by reducing net latent heating aloft and cooling below**. A kinematic definition would refer to this reduced precipitation and increased turbulence as invigoration. The authors' definition would refer to this as the opposite of invigoration. Some discussion of this is needed.

We have added to the introduction

Further, unlike studies that focus on the suppression of drizzle in shallow warm clouds, such as Ackerman et al. 2004 which found increased turbulence through suppression of drizzle by aerosol, herein we evaluate the effects of aerosol on warm rain events and define invigoration beyond just an increased in turbulence or vertical motion, but by changes in the latent heating structure throughout the cloud layer.

Most modeling studies that I know of suggest precipitation suppression in shallow marine cumulus, or at least a microphysical suppression that may then lead to a PBL/cloud deepening and a precipitation rate that is similar to the unperturbed state (e.g. Stevens and Seifert 2008). Can the authors provide some modeling studies that demonstrate increased precipitation with increasing aerosols?

We have added to the introduction, while discussing other modeling studies:

Recently, Wu et al. 2021 found in the Weather Research and Forecast model an increase in drizzle rates in the most polluted runs while simulating north Pacific warm clouds. Dagan et al. 2017 found that as clouds reach an equilibrium state, the polluted scenarios are likely to see an increase in rain production due to enhanced instability. Precipitation suppression by aerosol can also alter which type of clouds may eventually rain by altering water vapor transport, resulting in higher rain rates in regions downstream of the original suppression.

More detail is needed on the derivation of vertical motion profiles from the observations. The paper cited in the manuscript (Nelson et al., 2016) does not discuss how vertical motion profiles are derived. If there is no existing manuscript describing the methodology, then it needs to be described in this manuscript. There seems to be a major issue in my view in being able to infer both latent heating profiles AND vertical motion given the limited information (PIA plus the Z profile). The authors need to demonstrate that there is skill in this derived quantity.

The abstract states that the manuscript shows that cloud top entrainment rates are increased in response to aerosol. As with vertical motion profiles, I don't see how entrainment rates are derived with the available observations. Cloud top entrainment is a particularly challenging observation to make, and I suspect that changes in entrainment can only be inferred indirectly. Please clarify.

We apologize for the lack of clarity. We do not infer the direct entrainment, but the cooling due to cloud top evaporation. We have added to section 2.3
We refer to the evaporation at cloud top as due to entrainment, however WALRUS does not simulate entrainment rates, therefore we are inferring from the evaporation at the top of the cloud profiles that this cooling is due to entrainment.

The authors need to be precise in clarifying how invigoration is defined. The abstract introduces three separate metrics (precipitation, vertical motion, entrainment rate). I assume invigoration pertains to stronger updrafts under polluted conditions, not stronger precipitation rates. Is that correct? Line 182 suggests that it is something other than precipitation, i.e., turbulence, that they are referring to when they refer to invigoration.
We have added to the abstract
...by investigating the effects of aerosol loading on the latent heating and vertical motion profiles of warm rain.

Line 30: I'm confused. Which figure in Kubar et al., (2009) shows this? Kubar shows that deeper clouds precipitate more frequently and that precipitating clouds tend to have lower droplet concentrations. But I don't think it shows that clouds are deeper in polluted environments. Please provide supporting evidence.
We have corrected the statement. Kubar et al. 2009 said "For a given cloud-top height, however, warm clouds in more polluted regions tend to have more cloud liquid water than pristine cloud" and "We have seen that the macrophysical variables of cloud-top height and LWP are closely related, which is expected if cloud base is nearly constant." We have corrected line 30 to read: "Kubar et al. 2009 found evidence of increased liquid water amounts in highly polluted environments when controlling for cloud top height' instead.

Minor comment replies:

Line 8: What is the "pristine cloud response" responding to? I thought that the responses being investigated here are to aerosol, so I am a little confused.
 Added those with minimal anthropogenic aerosol emissions

Line 40. Jiang et al., (2009). Presumably the additional cooling results in stronger downdrafts. Is that the case, or are updrafts also enhanced?
Yes as stated, Jiang found an increase in vertical motion, meaning both up- and donwdrafts.

Line 42: What is "droplet mobility"?
Added: …the amount of motion by each droplet not forced by gravity…
From Koren et al 2015 "Here we study the mobility of cloud droplets in air. We use the term mobility to estimate how well droplets move together with the surrounding air as opposed to the deviation downward by gravity"

Line 45: "cloud lifetime with increasing lifetime"? Do you mean "cloud lifetime with increasing droplet concentration"?

Yes thank you for catching this typo. Corrected.

Line 53: This makes sense, because FT humidity tends to increase cloud longevity (see Eastman and Wood 2018, for example).

Added citation for Eastman and Wood 2018 to this example.

Line 61: Please motivate the choice of 150-200 g/m2 for the LWP range used here. Are the results at all sensitive to this choice?

This work in part builds on prior work (Douglas & L'Ecuyer 2019, 2020) which used a set of LWP constraints to quantify AIE. Added:

…,building on work by Douglas & L'Ecuyer 2019 and Douglas & L'Ecuyer 2020 which found this LWP range to be an inflection point for cloud lifetime effects.

Line 87: It would be useful to know what the distribution of cloud horizontal size looks like. AMSR-E footprints are variable depending upon the frequency, so 15 contiguous CloudSat pixels (~30 km) equates to what frequency of AMSR-E? In the subsequent figures (e.g. Fig. 2), why are results only shown for clouds up to 7 km in size?

After 7 km, it is less likely that the precipitation shown is for shallow convection clouds and more likely found in marine stratocumulus or marine stratocumulus transitioning to cumulus regimes. We have added that this work builds on prior work (2019 and 2020 papers), both of which explain the collocation techniques used to build the dataset.

Line 109. I assume that the latent heating rate is the difference between condensation and evaporation derived from the LES simulations. Can this be stated explicitly?

Added to manuscript.

Line 117: These simulations are not really LES, because their resolution is too coarse (250 m and 100 m vertical will not accurately represent the scales of mixing responsible for cloud top entrainment). I wonder how sensitive the LH profiles are to the vertical and horizontal resolution in the model. Has this been tested? The ATEX simulation used is Cu under Sc (cloud cover ~50%).

Currently this has not been tested as work on altering or testing WALRUS remains unfunded.

Is Figure 1 a single case as opposed to a composite of many cases? Please provide date, time and location if the former.

These are single cases from January 2007. Added to manuscript.

Line 127. "which include cooling by evaporation". What else do the cooling rates include?

Fixed to clarify, now reads "The maximum above cloud cooling due to evaporation is found by taking the maximum of all evaporative cooling rates starting at the cloud top to the top of the profile."

Line 128: Isn't the entrainment zone everywhere where there is latent cooling, not just where the profile shifts from positive to negative?
Yes, we use the cloud top to denote a rough start of the entrainment zone, however because we do not have the LH due to cooling and warming at each pixel, only the total LH, we can only roughly define the entrainment zone using cloud top height (which is usually where the profile switches from positive to negative LH).

There needs to be some attempt at quantifying the sampling uncertainties in Fig. 2. There needs to be a confirmation that there is no correlation between aerosol loading and stability in the different quadrants used (stable/unstable, polluted/clean). If there is, then the whole result could simply reflect meteorological covariation.
There will always be some correlation between aerosol and meteorology, as meteorology leads to certain aerosol conditions. We have added a note on this covariation, as well as the fact that even with our constraints, meteorology may act as a confounding variable.
Added:
> While our analysis does account for some amount of covariation between meteorology and aerosol-cloud interactions, there is some added uncertainty due to the inherent relationships between aerosol and meteorology, as certain meteorological conditions may lead to high or low aerosol loading.

Fig 2 caption. This is showing results as a function of the cloud size, so why does the caption state that the results are for clouds with an extent of 15 km?
We have limited the cross track cloud size to 15 km, and the x-axis of these plots is the size of the rain system within these clouds.

Line 149: I don't see an inflection point. I see a maximum at 5km and then a decrease. What am I missing? What do the authors mean by an inflection point? It is defined as where a function changes curvature. Do the authors mean a maximum?
We have altered the language to show that we mean the maximum, and that by inflection point, we mean "...whereby the behavior changes from increasing with rain size to decreasing…".

Lines 153:160. This paragraph is difficult to understand and read. First, I don't see how Fig. 3 shows what is being discussed. It doesn't show values in the center, but simply the maximum values. How do we know these are in the center? Does this mean the center in the horizontal or the vertical direction? Second, how is it possible to determine an inflection point at a size of 7km, when 7 km is the maximum size shown?
Altered wording for clarity, this paragraph now reads:
Polluted clouds exhibit first an increasing, then decreasing relationship between the size of the rain system and the maximum latent heating within the cloud; while pristine clouds show a steady, linear increase in the rain formation rate as the size of the rain system increases (Figure 3). Rain formation in polluted clouds appears to increase with rain size up to 5 km, then either decrease (unstable, dry) or remain constant with size. This inflection point, whereby the

behavior changes from increasing with rain size to decreasing, depends on both the stability of the boundary layer and the humidity of the free atmosphere. Pristine conditions do not show this same pattern, as for all meteorological conditions, an increase in rain size results in an increase in maximum heating.

Line 162: Should not start a new section with "However,…"
Removed however.

Line 165: How can evaporation take place above a cloud, when it is the cloud that is evaporating? I think you mean greater evaporation rates near cloud top. Also, cumulus clouds evaporate mostly by lateral rather than cloud top entrainment, so how is this factored into the analysis?
WALRUS only recreates a latent heating where there is rain certain according to the CloudSat profile. Lateral entrainment at the cloud edges would only be captured if the rain size was the same as the cloud size and the edges of the cloud profile were raining. It is unlikely that with our partitioning we included lateral entrainment. We have added as a caveat:
"Cumuli generally have large rates of lateral entrainment that would not be captured by WALRUS, however lateral entrainment would also affect the invigoration of any rain formation in the cloud layer."

Line 166: Under what conditions does increased mixing with the free atmosphere lead to cloud deepening? Mixing causes evaporation and a loss of buoyancy, and momentum friction, all of which slow the rise rate down. Exactly the opposite of what is stated here.
Added: "when the free atmosphere the boundary layer is unstable"

3: Why are the latent heating rates here so much higher than those in Fig. 2? Does one infer that nearly all the condensate evaporates?
These are the maximum within the center of the profile LH rates, while others are averaged over the profile reducing the magnitude.

Line 174. I do not understand this sentence at all.
Reworded to: While clouds in stable environments have similar responses in precipitation formation rates, inversion strength acts to limit evaporation at the cloud top.

Line 176: But the authors have essentially limited their cloud thickness by using only a narrow range of LWP values. So how can this deepening effect be determined if the cloud thickness is fixed?
Have added in theory as we do not investigate the cloud deepening response and this is a hypothetical analysis of what could lead to deepening responses seen by others.

Line 184: Cooling below cloud increases stability of the boundary layer unless the cooling is focused just below cloud. Often the cooling profile below cloud will maximize further down where it is relatively drier and thus promotes stability. This all depends upon the precipitation drop size. Here it is stated as decreasing stability as a general effect.

We have based our interpretation on Dagan et al. 2016 which focused on the below cloud evaporation. As WALRUS would capture below cloud cooling right below cloud base, and not at or near the surface, we believe our interpretation of the effects of below cloud evaporation are correct. However, we have added as a caveat to this:
"The effects of below cloud evaporation on the stability are sensitive to the location of the cooling and drop size of the cloud; it is possible in some circumstances that cooling can act to stabilize the boundary layer."

Line 210: This argument about stabilizing the PBL is the exact opposite of the one on Line 184, where the authors argued that evaporation below cloud can destabilize the PBL.
In unstable conditions, where there is a large difference in the heating higher in the cloud layer and cooling below, it can act to stabilize the boundary layer. Have altered the wording so that it is clear we only mean in certain conditions where the heating/cooling follows this pattern.

Line 217: How are the results consistent with this? Latent heat release is a consequence of precipitation and does not necessarily imply anything about turbulence as far as I can tell.
We have corrected the use of turbulence and vertical motion.

6: I don't understand how cloud updraft speeds of cm/s can be measured.
For more information on how WALRUS derives an updraft speed please see Nelson et al. 2016.

Line 246: Clarify what the difference is between peak and core.
Added clarification.

Line 249: They show the opposite of invigoration, which is suppression. This is not a dampening of invigoration. Again, there is confusion about the definition of invigoration. This sentence states that invigoration is dampened by reducing precipitation formation rates.
Have changed the language to inhibit, rather than damp, as damp may imply some amount of invigoration.

Line 256: "As the rain system grows, ..... are a function of the size of the rain system".
Grammatically it would be better to remove one of these references to size.
Changed the wording to reduce wordiness.

Line 265-266. This is not a sentence. I suggest putting in a semi-colon instead of the comma before "only".
Thank you for pointing out the structure of this sentence, reworked to clarify and make it grammatically correct.